# Photochemistry of the Cloud Aqueous Phase: A Review

**DOI:** 10.3390/molecules25020423

**Published:** 2020-01-20

**Authors:** Angelica Bianco, Monica Passananti, Marcello Brigante, Gilles Mailhot

**Affiliations:** 1Institute for Atmospheric and Earth System Research, University of Helsinki, 00014 Helsinki, Finland; angelica.bianco@helsinki.fi (A.B.); monica.passananti@helsinki.fi (M.P.); 2Dipartimento di Chimica, Università di Torino, 10125 Torino, Italy; 3CNRS, SIGMA Clermont, Institut de Chimie de Clermont-Ferrand, Université Clermont Auvergne, F-63000 Clermont-Ferrand, France; marcello.brigante@uca.fr

**Keywords:** atmosphere, photochemistry, cloud, oxidant capacity, hydroxyl radical

## Abstract

This review paper describes briefly the cloud aqueous phase composition and deeply its reactivity in the dark and mainly under solar radiation. The role of the main oxidants (hydrogen peroxide, nitrate radical, and hydroxyl radical) is presented with a focus on the hydroxyl radical, which drives the oxidation capacity during the day. Its sources in the aqueous phase, mainly through photochemical mechanisms with H_2_O_2_, iron complexes, or nitrate/nitrite ions, are presented in detail. The formation rate of hydroxyl radical and its steady state concentration evaluated by different authors are listed and compared. Finally, a paragraph is also dedicated to the sinks and the reactivity of the HO^•^ radical with the main compounds found in the cloud aqueous phase. This review presents an assessment of the reactivity in the cloud aqueous phase and shows the significant potential impact that this medium can have on the chemistry of the atmosphere and more generally on the climate.

## 1. Introduction

The role of clouds in the chemical composition of the atmosphere is mostly unknown. It affects the atmospheric oxidation capacity and global warming, which are critical in assessing air quality and regional climate. Between the different oxidizing species in the atmosphere, radicals play a central role in the cloud oxidation capacity, with a relevant impact on the chemical composition of the atmosphere [1].

Earth atmosphere contains a huge number of compounds that come from many and varied sources and whose concentrations are highly variable in space and time [2,3,4]. These compounds have scientific interest for many reasons: (1) they represent a serious health risk [5,6] and the World Health Organization (WHO) reports 90% of the population now breathe polluted air, which kills seven million people every year [7]; (2) they modify atmosphere oxidation capacity by their chemical and photochemical reactivity [4]; and (3) they modify aerosol properties and play a fundamental role in climatic change, by the fact that aerosols interact directly (effective radiative forcing aerosol radiation interaction (ERFari)) and indirectly (effective radiative forcing aerosol cloud interaction (ERFaci)) with solar radiations [8].

Biogenic and anthropogenic sources release in the atmosphere organic compounds as gases, vapors (volatile organic compounds (VOC)), or as aerosols. In the atmosphere, VOC oxidation leads to the formation of compounds with lower volatility. The latter, by nucleation and new particle formation [9,10,11], can form secondary organic aerosols (SOAs). SOAs represent a major concern in atmospheric chemistry research due to their abundance (over 50% of boundary layer aerosol particles are born via this process) and impact on climate and human health [12,13].

The atmosphere is a complex multiphase medium where gases, the solid phase (particles and ice crystals), and the liquid phase (cloud and fog droplets) coexist and interact. Solubility and Henry’s law determine the dissolution of gases in the liquid phase and, in particular, in cloud droplets. Deviation to this law could be because the droplets’ surface is not planar and hydrophobic organic matter or surfactants can be present at the air/water interface. When the cloud forms, aerosol particles can be englobed in cloud droplets and water-soluble organic carbon (WSOC) dissolves. Due to the numerous sources and number of soluble compounds from the gaseous and particulate phases, the aqueous phase of the cloud consists of a multitude of compounds. Currently, studies show that only 10%–30% of the dissolved organic matter in cloud water is characterized [14,15,16,17]. The most studied classes of compounds are short-chain carboxylic acids and carbonyl compounds (aldehydes and ketones). The aqueous phase of clouds also contains more complex organic compounds of anthropogenic or biogenic origin. These compounds are difficult to characterize but can influence the overall chemical composition of cloud droplets with an effect on the reactivity and toxicity of the cloud environment.

During the cloud lifetime, microphysical processes (like evaporation, condensation, collision, and coalescence) redistribute compounds between the phases and perturb chemical reactivity. This means that each cloud is a dynamic and multiphasic medium where chemical and photochemical reactions take place. Each cloud droplet is a small reactor and reactivity in the aqueous phase follows different pathways or, at least, different rates, than reactivity in the gaseous phase [18,19]. Reactions in cloud water are faster than in the gas phase and, consequently, chemical processes in cloud conditions are more relevant than in clear sky. Moreover, these chemical processes involve ionic species or they can happen at the air/water interface (heterogeneous reactivity), leading to produced compounds that are different from the ones produced in the gas phase. Composition and reactivity make clouds a specific environment, generally known as “cloud chemistry” [20,21,22,23,24,25].

## 2. Cloud Water Composition

Cloud water chemical composition depends on many variables and the reactivity is influenced by physicochemical parameters such as pH, temperature, and solar radiations. As reported previously, chemical compounds in cloud water come from the dissolution of aerosols [16,26,27,28]. Moreover, the water-condensed phase (liquid or solid) can dissolve gases as it allows the occurrence of reactions that would not otherwise happen or would be much slower in the gas phase.

Cloud water composition is directly related to its air mass origin, and it largely depends on the availability of soluble ionic species. In Figure 1 are reported the concentration of main ions as a function of the air mass origin. The fact that cloud droplets are formed on cloud condensation nuclei (CCN) has immediate implications for cloud water composition. In general, the fractional uptake of soluble aerosol species into cloud water is fairly high [29].

The main chemical species in cloud water are inorganic ions (such as Cl^−^, Na^+^, SO_4_^2−^, NO_3_^−^, NH_4_^+^, etc.) [14,24,30,31,32,33,34,35,36,37,38], transition metals (Fe, Cu, Mn, …) [26,38,39,40,41,42,43], and organic compounds (carboxylic acids, aldehydes, amino acids, …) [17,21,32,44,45,46,47].

The study of the chemical composition of cloud water droplets assumes fundamental importance in understanding its variability in the function of environmental conditions. Many campaigns of measurements on different sites were organized and the inorganic composition of cloud water was largely studied [14,24,26,27,32,36,37,48]. Figure 1 reports the median values of the concentration of main inorganic ions and pH for different sampling sites and air mass origins. The chemical compounds reactivity is then studied in the laboratory [49,50,51,52,53] and these data are useful for the development of atmospheric cloud water models for the prediction of the behavior of chemicals in such a complex medium.

### Physicochemical Parameters

The liquid water content (LWC) of clouds is one of the main controls of chemical concentrations in cloud water. The concentration of a cloud water solute is proportional to its concentration in the air and inversely proportional to LWC [54]. However, the cloud environment is very complex and other factors tend to distort this relationship—these factors and their impact on chemical concentration in cloud water are still under discussion [55].

Ionic species in cloud water are responsible for the conductivity of this medium and the total conductivity of the solution is expressed as the sum of the partial conductivity of each ion. This measurement allows estimating the ionic concentration of cloud water: the range of values measured by Cini et al. is 47–485 µS cm^−1^ in Vallombrosa (Tuscan Appennins, Italy) [39], the one from Kawamura et al. is 6–190 µS cm^−1^ in Los Angeles basin (US) [56], and Deguillaume et al. measured 2–348 µS cm^−1^ at Puy de Dôme summit (France) [14]. Higher values, up to 850 µS cm^−1^, were found in Taiwan during polluted cloud events [57]. Ionic species that mainly influence conductivity are H_3_O^+^, HO^−^ and then SO_4_^2−^, Ca^2+^, and other inorganic ions. 

One of the main factors influencing mass transfer from gas to droplets and cloud water reactivity is pH. The equilibrium value, considering CO_2_ concentration in air, is 5.65, but fluctuations are due to the solubilization of other species coming from gaseous (SO_2_, H_2_SO_3_, HNO_3_, NH_3_, carboxylic acids) or particulate phases (CaCO_3_). The simultaneous presence of many acid and basic compounds, such as nitrates, sulfate, ammonium, carbonates, and many organic acids, such as acetate and formate, leads to a buffer effect. Generally, the pH value is lower for air masses influenced by anthropic emissions caused by the solubilization of SO_2_, HNO_3_ and, carboxylic acids from the gas or the particulate phase [58,59]. Measured pH values are between 2.5 and 7.6 [14,37,60] and in most cases, a correlation between measured pH and air mass origin is observed, even if Budhavant et al. found pH values up to 7.4 for polluted air masses [61] probably due to the buffer activity of soil-derived calcium carbonate.

## 3. Cloud Aqueous Phase Reactivity

Chemical reactions of cloud droplets can take place at the interface and in the bulk and both should be taken into account. The main chemical transformations are initiated by free radicals or oxidant species, which can be generated in the presence of light or dark conditions. These reactions are difficult to study and model due to the extreme complexity of the matrix. Laboratory experiments are often limited to studies in the liquid phase, without considering the role of droplets’ surface and the equilibrium with the gas phase. Moreover, the complexity of the matrix is strongly simplified to allow the measurements in controlled laboratory conditions. However, the complexity of the matrix is particularly important for modeling and two kinds of models are currently developed:Implicit models, with simplified reactivity and empirical relationships, to give a quick and direct glance of dissolved organic matter or aerosol transformations;Explicit models, with a detailed description of the reactivity. These models are able to give precise degradation and formation pathways of organic compounds and radicals but they are only available for some compounds. 

Both approaches are valid and the specific model is chosen based on the scientific question. To study the reactivity of cloud waters, current models and experimental studies apply simplification to overcome the complexity of cloud droplets.

In this work, the attention is focused on the photoinduced formation of radicals and oxidant species from different precursors: these processes, due to solar radiations, represent the main production pathway of oxidant species. Their importance decreases during the night, when dark reactions become more important.

Although dark reactions are not the main focus of this work, they are briefly introduced in the next paragraph to better understand their importance in the real environment and their possible occurrence also during the day.

### 3.1. Reactivity in the Dark

In the absence of light, the main reaction pathways in cloud water are Fenton reaction and oxidation by ozone (O_3_). Fenton reaction leads to the oxidation of organic compounds by the formation of HO^•^ through the reaction R1.
(R1)Fe^2+^ + H_2_O_2_
→ Fe^3+^ + HO^•^ + HO^−^

The Fenton reaction can contribute, significantly, to the production of HO^•^, although the reactivity constant is relatively low (about 60 M^−1^ s^−1^ at 25 °C). However, the importance of the Fenton reaction regarding the production of HO^•^ in solution is still subject to controversy. The HO^•^ radical production by the Fenton reaction has been questioned by several studies that suggest that the reaction between H_2_O_2_ and Fe^2+^ produces the ferryl ion (Fe^4+^), which is then the active intermediate species in Fenton chemistry [62,63].

Ozone plays a central role in tropospheric chemistry: it is a highly reactive and toxic substance and it absorbs both ultraviolet and infrared light. Even in dark conditions, O_3_ can oxidize, as reported for the atmospheric oxidation of polycyclic aromatic hydrocarbons (PAH) [64,65]. O_3_ does not diffuse in cloud water and reactivity occurs mainly on the surface of the droplet. Furthermore, because O_3_ molecules are not formed in the aqueous phase, their chemical destruction rate per surface unit is equal to the net transfer flux of O_3_ from the gas phase. As a result, the washout of O_3_ is irreversible [66,67]. Concerning the surface chemistry of cloud droplets, only a little information is available [68].

### 3.2. Photochemical Reactivity

Organic and inorganic compounds can be transformed following two kind of photochemical processes. The first one is the direct absorption of solar radiation (photolysis) and second due to the oxidation mediated by photochemical generated species such as triplet states and radicals (photosensitized reactions).

#### 3.2.1. Photolysis

Photolysis is observed when molecules, after absorbing light, undergo a chemical structure change. In order to induce direct photolysis, the energy absorbed by the molecule should be higher than the energy of a covalent bond (usually in the range from 210 to 630 kJ/mol); therefore, the energy has to be generally higher than 210 kJ mol^−1^. The radiations containing this amount of energy are essentially the UV radiations, which are almost completely absorbed by O_3_ in the stratosphere. The quantity of UV radiation reaching the lower troposphere and the surface of the Earth is very small compared to the incoming radiation at the top of the atmosphere, but this small amount of UV radiation is the driving force for most of the photochemical reactions in the troposphere [69].

Photolysis has been studied in detail for pyruvate [70] and tryptophan [47] in synthetic cloud water, using different analytical techniques. The results showed that these compounds, that are able to absorb the UV portion of sunlight, are oxidized to smaller molecules, like lactate, acetate, and formate, or can form higher weigh molecular compounds, also at low concentrations, as showed for tryptophan. To our knowledge, direct photolysis for many compounds is negligible in comparison to reaction with oxidants.

#### 3.2.2. Photosensitized Reactions

Absorption of sun radiation can leads to the formation of excited state molecules that are able to start new photosensitized reactions [71]. Different kind of reaction can be initiated from the electronically excited state such as energy and electron transfer reactions. Despite a well knowledge of such reactions in surface water only limited work are reported in cloud waters [71].

Excited state organic compounds can react with molecular oxygen (^3^O_2_) leading to the formation of singlet oxygen (^1^O_2_). However, reactivity of ^1^O_2_ with cloud water relevant compounds is far to be assessed. In the condensed phase, electron transfer or H-transfer often follows the photochemical excitation of a chromophore: this kind of indirect photolysis is theoretically different from photosensitization and energy transfer, even if it could be a simultaneous process, and can lead to the production of neutral and charged radicals. For example, in the case of oxygen, it can form the superoxide radical anion (O_2_^•−^).

### 3.3. Photochemistry in Cloud Water

The fate of numerous organic and inorganic compounds in the atmosphere is controlled by photochemically produced oxidants or photooxidants. Historically, the field of atmospheric chemistry has focused on gas phase oxidations as the primary fate-determining processes of chemical substances. However, aqueous phase photochemical transformations in atmospheric water droplets can significantly affect the chemical composition of the troposphere [72]. Cloud droplets can undergo chemical changes through photochemical reactions because they receive a considerable amount of sunlight. Oxidation of chemical compounds in the aqueous phase is induced by radicals or other oxidant species, like O_3_ or H_2_O_2_ [73]. Ionic radicals (Cl_2_^•−^, Br_2_^•−^, SO_3_^•−^, SO_4_^•−^, SO_5_^•−^, and O_2_^•−^) are produced only in the aqueous phase, while neutral radicals, like HO^•^, NO_2_^•^ (nitrite radical), NO_3_^•^ (nitrate radical) and HO_2_^•^, can diffuse from the gas phase (gas to liquid transfer) or can be produced directly in the aqueous phase. Cited radicals do not have equal oxidation power: the redox potential increases following the order Br_2_^•−^ < Cl_2_^•−^ < NO_3_^•^ ≈ SO_4_^•−^ < HO^•^ [74]. Huie et al. reported a redox potential of 2.7 V for HO^•^, and 2.3 and 2.4 V for NO_3_^•^ and SO_4_^•−^, respectively [75]. Br_2_^•−^ has a negligible impact due to its low redox potential (1.69 V [75]) and low concentration, while SO_4_^•−^ and Cl_2_^•−^ recently gained more attention [76,77,78].

HO^•^ is the main oxidant in the atmosphere, especially during the day, and it is mostly responsible for transforming organic compounds in cloud water [79].

### 3.4. Main Oxidants in Cloud Water

The main oxidant species driving the cloud water oxidation capacity are H_2_O_2_, NO_3_^•^, and HO^•^. Between these species, HO^•^ is the most important oxidant during the day [80] and its formation and reactivity will be described in a separate chapter.

#### 3.4.1. Hydrogen Peroxide

##### Sources

H_2_O_2_ is produced in the gas phase and, in the presence of cloud droplets, rapidly dissolved in the liquid phase due to its high solubility. Modeling studies consider the mass transfer from the gas to the aqueous phase as the main source of H_2_O_2_ in cloud water [81,82]. Besides, several mechanisms have been proposed for the H_2_O_2_ photoproduction in atmospheric water droplets but only are based on laboratory experiments (R2–6). For example, the oxidation of transition metals ions (TMI) by radical species leads to the formation of H_2_O_2_ (R7–9): this is the case of the iron–oxalate complex, which is usually used as a model to describe the reaction of organic iron complexes in the cloud aqueous phase [83]. Even if the role of this complex is the subject of debate, the irradiation of cloud water leads to an increase in the concentration of H_2_O_2_ and Fe(II) [84]. The photolysis of phenolic compounds [85] and biacetyl compounds [86] was also proposed as a source of H_2_O_2_. Zuo and Deng observed that substantial amounts of H_2_O_2_ were produced by lightning activities during thunderstorms [87]. Another production pathway is the photolysis of organic peroxides, in particular of the methyl hydroperoxide, normally present in water droplets [53,88,89,90]. This photolysis in the aqueous phase is a source of HO^•^, which leads to the formation of formaldehyde and HO_2_^•^, which are sources of H_2_O_2_ (R10–11).
(R2)HO_2_^•^ + HO_2_^•^→ H_2_O_2_ + O_2_k_11_ = 8.3 × 10^5^ M^−1^s^−1^ *(R3)HO_2_^•^ + H_2_O → (HO_2_^•^−H_2_O)k_12_ = 9.7 × 10^7^ M^−1^s^−1^ *(R4)HO_2_^•^ + (HO_2_^•^−H_2_O) → H_2_O_2_ + O_2_ + H_2_Ok_13_ = 9.6 × 10^7^ M^−1^s^−1^ *(R5)(HO_2_^•^−H_2_O) + (HO_2_^•^−H_2_O) → H_2_O_2_ + O_2_ + 2H_2_Ok_14_ = 9.6 × 10^7^ M^−1^s^−1^ *(R6)HO_2_^•^
⇄ O_2_^•−^ + H^+^pk_a_ = 4.88(R7)Fe(II) + O_2_^•−^ + 2H^+^
→ Fe(III) + H_2_O_2_k_16_ = 1.2 ÷ 2.1×10^6^ M^−1^s^−1^ *(R8)Fe(II) + O_2_ → Fe(III) + O_2_^•−^pH dependent(R9)O_2_^•−^ + HO_2_^•^+ H_2_O → H_2_O_2_ + O_2_ + HO^−^k_18_ = 9.7 × 10^7^ M^−1^s^−1^ *(R10)CH_3_OOH + hν → CH_3_O^•^ + HO^•^
(R11)CH_3_O^•^ + O_2_
→ H_2_CO + HO_2_^•^
* Constants from the National Institute of Standard and Technology (NIST) Solution Kinetics Database.

##### Steady State Concentration

H_2_O_2_ concentration in cloud water depends on many factors such as air mass origin, season, and location of sample site. Previous measurements at the Puy de Dôme station show that H_2_O_2_ concentrations in the cloud aqueous phase range from 0.3 to 58 µM [14,91], while a study on cloud water sampled in Los Angeles shows concentrations up to 88 µM [92]. The concentration of H_2_O_2_ in cloud water collected at Mount Tai (China) was found to be up to 97 µM [38]. Table 1 reports the concentration of H_2_O_2_ measured on altitude sites.

##### Sinks in the Liquid Phase

The main removal mechanisms for H_2_O_2_ in the liquid phase are the photolysis (λ < 370 nm) (R12) [94] and reaction with HO^•^ (R13) [95].
(R12)H_2_O_2_ + hν → 2HO^•^
(R13)H_2_O_2_ + HO^•^
→ HO_2_^•^ + H_2_Ok_22_ = 3.2 × 10^7^ M^−1^s^−1^ ** Constants from NIST (National Institute of Standard and Technology) Solution Kinetics Database.

In the gas phase, photolysis leads to a significant loss of H_2_O_2_ in the troposphere (R12), although the absorption drops rapidly at a wavelength above the actinic cutoff of 290 nm. The quantum yield is of two HO^•^ for each H_2_O_2_, corresponding to a photodissociation quantum yield of 2 at wavelengths >222 nm [94]. In the condensed phase, the quantum yield is lower because the formed radicals have a higher probability of recombining due to the cage solvent effect. Moreover, in the cloud aqueous phase, H_2_O_2_ can be consumed by Fenton processes. Many other inorganic compounds have an impact on H_2_O_2_ degradation. Zuo and Deng found an inverse correlation between H_2_O_2_ and NO_3_^−^ and SO_4_^2−^ concentrations in rainwater and cloud waters [87]. This phenomenon is explained by the oxidation of sulfites (S(IV)) to sulfates (S(VI)) and of ammonium (NH_4_^+^) to nitrates, where the role of H_2_O_2_ (directly or as an HO^•^ source) is important at typical hydrometeor pH values [92,96,97,98].

#### 3.4.2. Nitrate Radical

##### Sources

The photolysis of nitrate ions leads to the formation of NO_2_^•^ and NO^•^ radicals (R14–15), however there is no evidence that nitrate radicals (NO_3_^•^) are formed by NO_3_^−^ irradiation [99,100]. NO_3_^•^ can be generated by the electron transfer between nitrate ions and aqueous radical anions, however the main source of nitrate radical in cloud water is due to mass transfer from the gas to the aqueous phase [101].
(R14)NO_3_^−^ + hν → NO_2_^•^ + O^•−^(R15)NO_2_^−^ + hν → NO^•^ + O^•−^

##### Steady State Concentration and Sinks

NO_3_^•^ absorbs strongly in the red region (620−670 nm) of the visible spectrum, unlike most atmospherically important species that absorb in the UV region. Due to this absorption, during the day, it gives photodissociation producing NO_2_^•^ or NO^•^ (R16–17).

The spectroscopic proprieties of NO_3_^•^ allow a good estimation of its concentration in the dry troposphere during the day (10^4^ molecules cm^−3^) and during the night (10^9^ molecules cm^−3^) [69]. In the case of wet air, NO_3_^•^ reacts with NO^•^ to give N_2_O_5_ that, in the presence of water, forms HNO_3_, responsible for acid fog and rain (sink of nitrate radical by wet deposition) [102]. The exchange of NO_3_^•^ with the aqueous phase was investigated by Thomas et al., [103] at room temperature (293 K). From these experiments, the uptake coefficient of NO_3_^•^ (γ(NO_3_^•^)) was found to be ≥2 × 10^−3^ while the Henry coefficient was estimated to be K_H(NO3•)_ = (3.8 ± 3) × 10^−2^ M atm^−1^ [104]. Because of its low solubility, the heterogeneous removal of NO_3_^•^ is only important when the dissolved NO_3_^•^ is removed quickly from the equilibrium, for example by reactions with Cl^−^ or HSO_3_^−^ ions (R18–19) in the liquid phase. Otherwise, heterogeneous removal should mainly proceed via N_2_O_5_ [103], with the production of HNO_3_ and consequent inactivation of this radical in the condensed phase.
(R16)NO_3_^•^ + hν < 635 nm → NO_2_^•^ + O(^1^D)
(R17)NO_3_^•^ + hν < 586 nm → NO^•^ + O_2_
(R18)NO_3_^•^ + Cl^−^
→ Cl^•^ + NO_3_^−^k_30_ = 1.0 ×10^7^ ÷ 1.0×10^8^ M^−1^s^−1^ *(R19)NO_3_^•^ + HSO_3_^−^
→ SO_3_^•−^ + NO_3_^−^ + H^+^k_31_ = 1.4 ÷ 2.0 × 10^9^ M^−1^s^−1^ ** Constants from the National Institute of Standard and Technology (NIST) Solution Kinetics Database.

## 4. Hydroxyl Radical

The main generation and destruction pathways of HxOy species are resumed in Figure 2 and described below.

HO^•^ drives the daytime chemistry of both polluted and clean atmosphere [80]. The HO^•^-mediated oxidation of organic compounds in the aqueous phase can lead to the formation of shorter but often multifunctional organic species and, ultimately, to complete mineralization. Complex chemical reactions triggered by HO^•^ can also occur in the aqueous phase forming accretion products such as oligomers [105]. These alternative chemical pathways are efficient processes to convert organic compounds into SOAs [106].

HO^•^, HO_2_^•^, and H_2_O_2_ are interconnected and grouped as H_x_O_y_. They have a central role in cloud water oxidation capacity [107]. As reported previously, they have high solubility [104] and the mass transfer from the gas to the aqueous phase can readily perturb gas phase reactivity [66,108,109]. HO^•^ can be produced in the gas phase and then diffuse to the liquid phase or it can be directly produced in the aqueous phase. On the contrary of what was previously described for the nitrate radical, HO^•^ can diffuse to the aqueous phase after its production in the gas phase (K_H(HO•)_ = 30 ± 0.2 M atm^−1^) [110]. In the following paragraph, the main sources are described.

### 4.1. Sources in the Aqueous Phase

The sources of HO^•^ in the aqueous phase strongly differ from those in the gas phase because of the presence of ionic species and metal ions. To our knowledge, little information is available in the literature concerning measurements of *R^f^_HO_*^•^ (HO^•^ formation rate) in real cloud water samples. Faust and Allen [111] measured the *R^f^_HO_*^•^ of six continental cloud water samples under monochromatic irradiation at 313 nm and they found values ranging from 1.3 to 8.3 × 10^−10^ M s^−1^. Anastasio and McGregor [112] investigated the photoreactivity of two cloud waters from the Tenerife Islands. The authors found *R^f^_HO_*^•^ ranging between 3.0 and 6.9 × 10^−10^ M s^−1^ and suggested that long-range terrestrial aerosol and gas transport in continental clouds could provide an additional source of HO^•^ compared with other marine or remote clouds. Bianco et al. [93] measured *R^f^_HO_*^•^ for 36 samples of marine and continental origin collected at the Puy de Dôme station and found values ranging between 0.2 and 4.2 × 10^−10^ M s^−1^. Similar results were found previously by Arakaki and Faust [84], which measured *R^f^_HO_*^•^ ranging between 0.3 and 5.9 × 10^−10^ M s^−1^.

Some work also focuses on the determination of the main precursor of HO^•^: Kaur and Anastasio [113] determined that the main source of HO^•^ in fog water is due to nitrate photolysis, while Bianco et al. [93] showed that 70% to 90% of HO^•^ in cloud water is due to H_2_O_2_. These results could seem to be in contrast but it should be considered that fog water samples collected by Kaur and Anastasio [113] in California and Louisiana (USA) were frozen before analysis and negative temperatures affect the concentration of H_2_O_2_. Moreover, the concentrations of nitrates were up to 1830 µM, one order of magnitude higher than the ones measured in cloud samples by Bianco et al. [93]. Furthermore, the emission spectrum of the lamp used to determine *R^f^_HO_*^•^ may affect the results because H_2_O_2_, nitrates, nitrites, and iron absorb light in different ranges. The following paragraphs report the main sources of HO^•^ in the aqueous phase.

#### 4.1.1. Hydrogen Peroxide

An important source is the photolysis of H_2_O_2_ (R12). Many authors measured the H_2_O_2_ quantum yield for the irradiation of solution at 254 nm, as reported by Herrmann et al. [79], while little and discordant information is reported for irradiation wavelengths higher than 300 nm and more representative of environmental conditions (Table 2).

#### 4.1.2. Iron Photochemistry

Iron plays a central role for evaluation of cloud water oxidant capacity because it is either a source or a sink of H_x_O_y_. In particular, it produces HO^•^ radicals by the Fenton reaction or iron photolysis. From an atmospheric point of view, iron is probably the most significant transition metal because of its concentration, which is, in general, much higher than that of other metals. Its concentration is ~10^−6^ M, but many field experiments indicate that it can vary from 10^−9^ to 10^−6^ M in raindrops and cloud droplets [28,39,40,43,118,119]. The concentration of iron drives the concentration of radicals in cloud droplets, but also partly in the gaseous phase (due to the rate of the mass transfer). In the cloud aqueous phase, the redox cycle between Fe(II) and Fe(III) depends on many factors such as pH, the concentration of oxidant, reducing and complexing agents, and the intensity of the actinic irradiation. The main chemical pathways driving the reactivity of the H_x_O_y_/iron system in cloud water are presented in Figure 3a. Iron(III) is present in three monomeric forms: Fe^3+^, Fe(OH)^2+^, and Fe(OH)_2_^+^. In cloud droplets, the speciation of iron between its two oxidation states (II and III) is a key parameter of its reactivity in solution and is a function of pH and redox potential, as shown in Figure 3b. Fe(II) is oxidized by H_2_O_2_ via the Fenton reaction (R21) to form Fe(III) and HO^•^. Under irradiation, this reaction is in competition with the photolysis of H_2_O_2_ (R12) and the photoreduction of Fe(III) (R20) [120]. Furthermore, as shown in Figure 3a, HO^•^ can oxidize iron (reaction D′) and react with H_2_O_2_ to form HO_2_^•^/ O_2_^•−^. These radicals can trigger the oxido-reduction of iron (reactions E and E′) and generate H_2_O_2_. Iron complexes, like the iron–oxalate complex, can also undergo Fenton reaction, as shown in reaction R22. The relevance of iron photochemistry in HO^•^ production is high, in particular for air masses of continental origin, but it still needs to be quantified.
(R20)Fe^(III)^(OH)^2+^ + hν → Fe^2+^ + HO^•^(R21)Fe^(II)^(OH)^+^ + H_2_O_2_
→ Fe(OH)^2+^ + HO^•^ + HO^−^(R22)Fe^(II)^(C_2_O_4_) + H_2_O_2_
→ Fe(C_2_O_4_)^+^ + HO^•^ + HO^−^

The Fenton reaction may significantly contribute to *R^f^_HO_*^•^, especially in the presence of Cu^+^, which can reduce Fe^3+^ to Fe^2+^, as reported in reaction R23.
(R23)Cu^+^ + Fe^3+^
→ Cu^2+^ + Fe^2+^k = 1.3 × 10^7^ M^−1^s^−1^ [121]

TMI interactions modify their oxidation state and impact H_x_O_y_ redox cycle. Tilgner et al., report that conversions between Fe^2+^/Fe^3+^, Cu^+^/Cu^2+^, and Mn^2+^/Mn^3+^ are efficient and represent sources and sinks for HO^•^ [122].

Another factor that influences iron chemistry is the formation of a complex with organic ligands [123,124], like oxalate, malonate, and tartronate [125]. Reactivity and photoreactivity of the Fe–oxalate complex (complexation constant log K = 9.4 [126]) have been deeply investigated in the past considering various complexation degrees (i.e., various Fe/oxalate ratios) [83,118], as well as the influence of pH [120]. Nowadays, the Fe–oxalate complex represents the main form considered in the cloud chemistry model [81,108,122], even if some other organic ligands start to be considered. Organic ligands produced by biological sources, like exopolymeric substances and siderophores, as well as humic-like substances (HULIS), show complexation constant with orders of magnitude higher than oxalate [4,15]. Notably, siderophores are proteins synthesized within the microbial cell and expelled to bind iron and make it bioavailable, since it is necessary as an active site in many enzymes [127]. Recently a screening work on 450 microbial strains isolated from cloud water evidenced that 43% are able to produce siderophores like pyoverdine [128] and the complexation constant is higher than 10^20^ [129]. A complementary work on the photoreactivity of the iron–pyoverdine complex reported that siderophores bind efficiently iron and reduce its HO^•^ quantum yield to one order in magnitude lower than values measured for Fe–oxalate complexes [130]. Since the complexation constant of Fe–oxalate is lower than the one for Fe-siderophores, these compounds may compete with oxalate in iron complexation and reduce its role in cloud photochemistry.

#### 4.1.3. Nitrate and Nitrite Photolysis

As discussed in Section 2, many studies report the concentration of nitrate in cloud water. Nitrate photolysis was firstly investigated in seawater [131]. Nitrate can absorb sunlight and their photolysis gives NO_2_^•^ and O^•−^ or O(^3^P) (ground state oxygen atom) (R14 and R24−26), which is responsible for the formation of HO^•^. NO_2_^•^ reacts following many pathways (R27−34) and produces nitrite anions.
(R24)NO_3_^−^ + hν → NO_2_^−^ + O(^3^P)Φ_(NO2−)305nm_ = 0.72 ÷ 1.22 × 10^−3^ [132]

Φ_(NO2−)313nm_ = 1.1 ± 0.2 × 10^−2^ [133](R25)O^•−^ + H_2_O → HO^−^ + HO^•^
(R26)O(^3^P) + H_2_O → 2HO^•^
(R27)NO_2_^•^ + HO^•^
→ HNO_3_
(R28)HNO_3_ + hν → NO_3_^−^ + H^+^
(R29)HNO_3_ + hν → NO_2_^•^ + HO^•^
(R30)NO_2_^•^ + O_2_^•−^
→ NO_2_^−^ + O_2_
(R31)2NO_2_^•^
→ N_2_O_4_
(R32)2NO_2_^•^ + H_2_O → NO_2_^−^ + NO_3_^−^ + 2H^+^
(R33)N_2_O_4_ + H_2_O → NO_2_^−^ + NO_3_^−^ + 2H^+^
(R34)NO_2_^−^ + HO^•^
→ NO_2_^•^ + HO^−^


Nitrous acid (HNO_2_) is a compound normally studied in the gaseous phase because it is strictly connected to the NO_x_ cycle. Cloud and fog droplets could be sinks of HNO_2_ from the gaseous phase because of its high value of Henry’s law constant (K_H(HNO__2)_ = 50 M atm^−1^). Nitrous acid in droplets can undergo photolysis (R35) but, more probably, it follows another reaction pathway: at water droplet pH values, nitrous acid is present in the deprotonated form, nitrite. Even if the concentration is very low (of the order of 0.1 µM), nitrites can absorb sunlight more efficiently than nitrates and they photogenerate HO^•^ following the reactions R36, R15, and R37.
(R35)HNO_2_ + hν → NO^•^ + HO^•^(R36)HNO_2_
⇄ H^+^ + NO_2_^−^ pk_a_ = 3.35(R37)O^•−^ + H_2_O → HO^−^ + HO^•^

### 4.2. Reactivity

The reaction of HO^•^ with ions is often described as a simple electron transfer (R38), but such a simple process is unlikely because of the large solvent reorganization energy involved in forming the hydrated hydroxide ion. Instead, it is suggested that an intermediate adduct is formed. Such an adduct is observed in the oxidation of halide and pseudo-halide ions (R39). Although there are several examples of HO^•^ reacting with inorganic ions at the diffusion-controlled rate, rate constants for the oxidation of many metal cations seem to be no more than ~3 × 10^8^ M^−1^s^−1^. A suggested reason for this is that HO^•^ abstracts H from a coordinated water molecule and this is followed by an electron transfer from the metal to the oxidized ligand (R40).

In a strongly alkaline solution, HO^•^ is rapidly converted into its conjugate base O^•−^, with a pKa value of 11.9. In its reactions with organic molecules, HO^•^ behaves as an electrophile whereas O^•−^ is a nucleophile and it is generally less reactive. This species is important at pH values higher than 12 and it will not be considered in this work.
(R38)HO^•^ + Fe^2+^
→ Fe^3+^ + HO^−^
(R39)HO^•^ + Cl^−^
⇄ HClO^•−^
(R40)HO^•^ + Fe^2+^
→ Fe^2+^HO^•^
→ Fe^3+^ + OH^−^


HO^•^ reacts by different pathways (double bond addition, hydrogen abstraction, electron transfer, and aromatic ring addition) with organic molecules, as schematized in Figure 4.

HO^•^-mediated oxidation of organic compounds can lead to the fragmentation or to the formation of oxidized organic species, introducing different functional group, to result in the complete mineralization to CO_2_. However, another pathway is possible: when water evaporates and solutes become more concentrated, the recombination of organic radicals becomes possible. The result is the formation of dimers, oligomers or, in general, large molecular compounds (LMC), as shown in Figure 4 (right). This phenomenon is still uncertain and can lead to the formation of oligomers up to 10 monomeric units [106]. The polymerization could also take place at the interface of the water droplets where the hydrophobic compounds can accumulate [134]. The competition between fragmentation and formation of high weight molecular compounds is well known for oxalic acid that, by reacting with HO^•^, can mineralize to carbon dioxide or dimerize [52].

HO^•^ reacts with organic compounds mainly by hydrogen abstraction or electron transfer, forming an alkyl radical R^•^. In the presence of dissolved oxygen, this latter reacts to form peroxyl radical RO_2_^•^ [135]. If peroxyl radical contains an alcoholic function (–OH) on α carbon, RO_2_^•^ rapidly decomposes producing HO_2_^•^/O_2_^•−^ [136,137]. RO_2_^•^ can also react by “self-reaction” with another peroxyl radical, R’O_2_^•^, with the formation of a link between the two oxygen atoms, giving a tetroxide ROOOOR’, highly unstable, which decompose quickly by different ways [138,139]. Tetroxide decomposition can lead to the formation of stable products, like aldehydes, ketones and alcohols [137,138], or radicals, as acylperoxides and alkoxylates. The latter reacts quickly by decarboxylation, electron transfer, or breaking of the C–C bond [140,141].

Phototransformation of carbonylic and carboxylic compounds has been largely investigated by laboratory studies by irradiation of natural or synthetic cloud water containing radical sources. The behavior of glyoxal [142], methylglyoxal [143], methacrolein, and methyl vinyl ketone [144,145], but also of carboxylic acids like formic, oxalic, acetic, glyoxylic, glycolic, pyruvic, and others [52], was investigated in synthetic cloud water to evaluate photodegradation pathways.

Recently, amino acids reactivity was investigated in cloud water: irradiation of tryptophan in the presence of H_2_O_2_ in synthetic cloud water evidences the production of acetic and formic acids and the degradation mechanism was investigated by mass spectrometry [47]. The photochemical reactivity of Fe(III)–amino acid complex in atmospheric waters was studied by Marion et al. [146]. The photolysis of Fe(III)–aspartate complex under sun-simulated conditions leads to the formation of ammonia as the main product and, as for tryptophan, to short-chain carboxylic acids. The authors demonstrated that the formation of iron–amino acid complexes could represent a degradation pathway for amino acids oxidation as well as new photochemical sources of carboxylic acids and ammonia in the cloud aqueous phase. The reactivity of hydroxyl radical with phenol, nitrophenols, and brown carbon has been studied in model cloud water and several studies pointed out the enhanced reactivity in the aqueous phase compared to the liquid phase. Moreover, Zhang et al. show an important result: the reaction between HO^•^ and brown carbon leads to an initial increase of the UV visible absorption followed by a rapid decrease (bleaching), with a potential impact on aerosol radiation interaction [147,148,149,150].

Although several studies investigated the fate of organic compounds in cloud water, the reactivity of a huge amount of organic species is still not considered in cloud water chemistry and photochemistry, mainly due to their low concentration and complex determination of degradation pathway.

### 4.3. Hydroxyl Radical Sinks

The HO^•^ is an important oxidant in the atmospheric aqueous phase, where it reacts with both organic and inorganic species. HO^•^ is scavenged in the aqueous phase, primarily by dissolved organic compounds: for example, the HO^•^-mediated oxidation of aqueous glyoxal, glyoxylic acid, and other small, multifunctional organic compounds produces low-volatile SOAs species, such as oxalic acid and oligomers. The lifetime of an aqueous species “S” considering the reactivity with HO^•^ is inversely proportional to the [HO^•^]_ss_ (HO^•^ steady state concentration), generally expressed by Equation (1):(1)$$\tau \;=\;\frac{1}{k_{S,HO^{•}}\times {\left[HO^{•}\right]}_{ss}}$$
where *k_S,HO•_* is the bimolecular rate constant between S and HO^•^. Thus, [HO^•^]_ss_ is a crucial quantity for understanding the fates of atmospheric pollutants. This concentration is determined by the balance of the HO^•^ sources and sinks, reported in Equation (2):(2)$${\left[HO^{•}\right]}_{ss}=\;\frac{P_{HO^{•}}}{k_{HO^{•}}^{\prime }}$$
where *P_HO•_* is the rate of production of HO^•^ and *k’_HO•_* is the apparent first-order rate constant for loss of HO^•^ (s^−1^). Since there are many scavengers for HO^•^, the rate constant for HO^•^ loss is the sum of the individual scavenger contributions and can be calculated, as reported in Equation (3), as the product of the bimolecular rate constant and the scavenger concentration for each species
(3)$$k_{HO^{•}}^{\prime }=\;\sum^{\;}(k_{S,HO^{•}}\times \left[S\right])$$

Determining the rate constant for HO^•^ loss in this manner is a herculean task since atmospheric drops and particles can contain on the order of 10^4^ individual organic compounds, as well as significant concentrations of poorly characterized, large molecular weight species such as humic-like substances [15]. In contrast, the most sophisticated numerical models of atmospheric waters track the reactions of fewer than 100 individual organic compounds, with none larger than four carbon atoms [81,122].

Arakaki et al. found that the HO^•^ sink can be simply estimated by a general carbon rate constant that is applicable both for atmospheric waters as well as surface waters, which allows *k’_HO•_* to be estimated by using organic carbon concentrations [151]. Their results show that the scavenging rate constant of HO^•^ by organic species in atmospheric waters can be simply estimated as the product of a robust general rate constant (*k_DOC,HO•_*) multiplied by the dissolved organic carbon concentration of the sample, as reported in Equation (4).
(4)$$k_{HO^{•}}^{\prime }=\;k_{DOC,HO^{•}}\times \left[DOC\right]$$

Equation (4): first-order rate constant for HO^•^ loss as a function of the general second order rate constant between HO^•^ and DOC.

In the work of Arakaki et al., *k_DOC,HO•_* is estimated to be of the order of 3.8 ± 1.9 × 10^8^ L molC^−1^ s^−1^ [151]. This value is useful for a direct estimation of the contribution of DOC as a scavenger of HO^•^.

### 4.4. Steady State Concentration

Uncertainties in HO^•^ sinks and sources make its concentrations in atmospheric water highly difficult to determine. To the best of our knowledge, only two works report the measurement of [HO^•^]_ss_, which is a crucial quantity to understand the fate of atmospheric pollutants [151] for cloud water samples: McGregor and Anastasio [112], who reported values ranging from 1.7 to 7.7 × 10^−16^ M, and Lallement et al., who measured [HO^•^]_ss_ of 7.2 ± 5.0 × 10^−16^ M. 

[HO^•^]_ss_ has been also estimated using cloud chemistry models. Contrarily to what happened for experimental measurements, cloud chemistry models are able to describe multiphase chemistry and mass transfer from the gaseous to the aqueous phase. [120,152,153]. The range of HO^•^ concentration varies from 10^−16^ to 10^−12^ M, depending on the “chemical scenario” (i.e., emission/deposition and the initial chemical conditions) used in the modelling study, as shown in Table 3, where the estimation of the [HO^•^]_ss_ for different air mass origins is reported. The amounts of organic matter and iron are key parameters controlling the [HO^•^]_ss_. These models are expected to underestimate the radical sinks because organic scavengers cannot be exhaustively described in the aqueous chemical mechanism [151]. Besides, they are able to consider the mass transfer of HO^•^ from the gaseous to the aqueous phase, which represent an additional source for [HO^•^]_ss_. For this reason, the measured value reported by Lallement et al. [154] and McGregor and Anastasio [112] is in the lower range of the evaluations reported by Tilgner et al. [122] using the CAPRAM 3.0 model.

## 5. Conclusions

This review highlights the complexity of the atmospheric aqueous phase in terms of a variety of chemical compounds present but also in terms of chemical processes taking place in this medium. The obtained results on the different studies demonstrate that photochemistry and more particularly the photogenerated hydroxyl radical drive the oxidation capacity during the day. Moreover, it is also clear that the atmospheric aqueous phase has a non-negligible impact on atmospheric chemistry and so on the climate. To evaluate more clearly the role and the significance of the atmospheric aqueous phase, more in-depth knowledge is needed both for the characterization of this aqueous phase and for the various processes that govern its composition. Its new data are also essential to establish more accurate atmospheric chemistry models to predict climate change, the level of atmospheric pollution, and so the atmospheric composition.

## Figures and Tables

**Figure 1 molecules-25-00423-f001:**
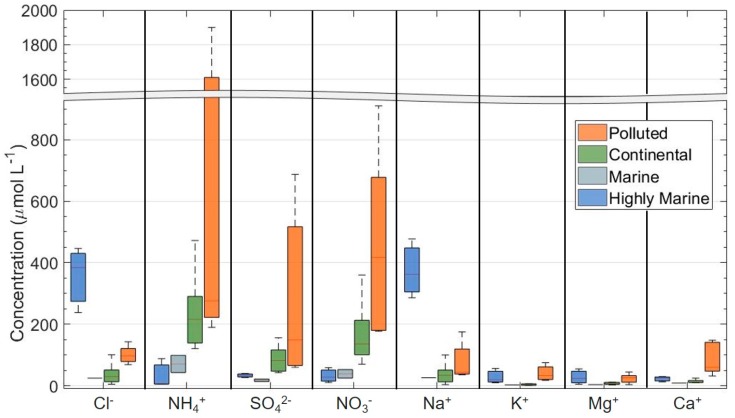
Concentration of the main ions in cloud water for different air mass origin (highly marine, marine, continental, and polluted) measured in nine different sites (Mt. Brocken, Germany; Vosges, France; Mt. Rax, Austria; Puy de Dôme, France; Szrenica, Poland; East Peak, Puerto Rico; Whiteface Mt., US; Mt. Tai, China; Mont Smücke, Germany). The full line represents the median values. The bottom and top lines correspond to the 25th and 75th percentiles, respectively. The ends of the whiskers are the 10th and 90th percentiles. Medians and percentiles are calculated on the average of the measurement reported in each work.

**Figure 2 molecules-25-00423-f002:**
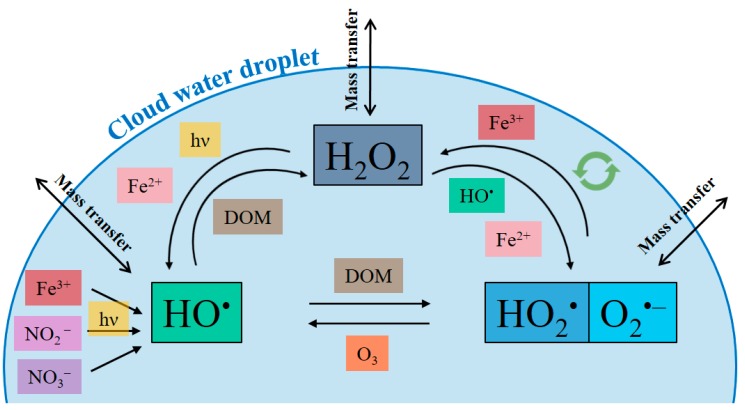
Main HO^•^ generation and destruction pathways. Double circular arrow (light green) means radical recombination. DOM (dissolved organic matter); VOC (volatile organic compounds); hν means light.

**Figure 3 molecules-25-00423-f003:**
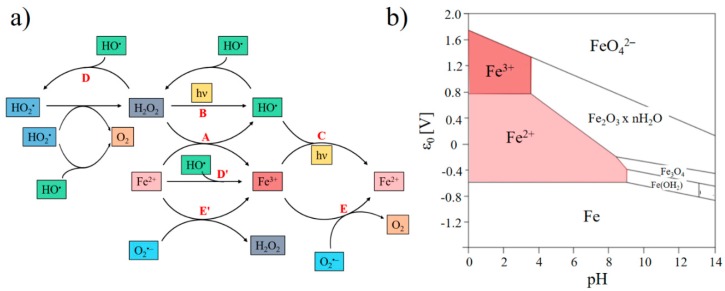
(**a**) Iron chemistry and photochemistry (adapted from Long et al. [120]); (**b**) Pourbaix diagram of iron: speciation as a function of pH and redox potential.

**Figure 4 molecules-25-00423-f004:**
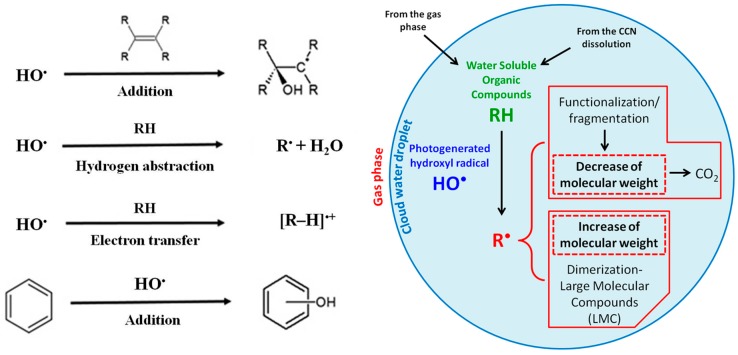
HO^•^ reaction pathway (**left**) and becoming of organic compounds after reaction (**right**).

**Table 1 molecules-25-00423-t001:** Average for H_2_O_2_ in cloud water sampled at different sites.

			H_2_O_2_ Average Concentration (μM)			
Location	Altitude (m)	AMO	Min	Moy	Max	Reference
Puy de Dôme, France	1465	P	1.9	4.9	7.3	[14]
Puy de Dôme, France	1465	C	1	9.9	57.7	[14]
Puy de Dôme, France	1465	M	0.1	6.2	20.8	[14]
Puy de Dôme, France	1465	HM	0.8	11.2	19	[14]
Mont Smücke, Germany	937	C	0.4	5.6	17	[37]
Puy de Dôme, France	1465	M	2.1	12.1	52.3	[93]
Mont Tai, China	1534	P	0	23.5	97.1	[38]

AMO, air mass origin; HM, highly marine; M, marine; C, continental; P, polluted.

**Table 2 molecules-25-00423-t002:** Summary of HO^•^ quantum yields (Φ_OH_) for H_2_O_2_ photolysis in aqueous solution at different photolysis wavelengths.

λ (nm)	Φ(HO^•^)	Reference
300	0.96 ± 0.09	[114]
308	0.98 ± 0.03	[79]
308	0.8 ± 0.2	[115]
313	0.98 ± 0.07	[116]
313	0.59 ± 0.01	[76]
334	0.98 ± 0.008	[116]
351	0.96 ± 0.04	[79]
360	0.017	[117]
365	0.96 ± 0.09	[114]
365	0.009 ± 0.001	[76]
400	0.96 ± 0.09	[114]

Some values are discordant but an average value close to 1 seems to reflect the reality of this very effective reaction.

**Table 3 molecules-25-00423-t003:** Calculated HO^•^ radical concentrations in clouds and deliquescent particles using the CAPRAM 3.0 multiphase mechanism. Mean concentrations are averaged values over three simulation days. Data adapted from Tilgner et al. [122].

Polluted Origin Cloud Water [HO^•^] (M)			Remote Origin Cloud Water [HO^•^] (M)			Marine Origin Cloud Water [HO^•^] (M)		
Mean	Max	Min	Mean	Max	Min	Mean	Max	Min
3.5 × 10^−15^	1.6 × 10^−14^	2.9 × 10^−16^	2.2 × 10^−14^	6.9 × 10^−14^	4.8 × 10^−15^	2.0 × 10^−12^	5.3 × 10^−12^	3.8 × 10^−14^

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
