# Peer review of "Photochemistry of the Cloud Aqueous Phase: A Review"

_molecules, 2020, doi:10.3390/molecules25020423_

Round 1
Reviewer 1 Report
This review summarizes the literature on radical chemistry in clouds at day- and nighttime.
It is interesting as a compilation of the literature, but does not provide original insights.
It points out to the complexities but makes no effort to indicate what simplifying assumptions could be applied to deal with complexity.
Fig. 3b, the Pourbaix diagram of iron speciation as function of pH is well known, but, as reported by the authors in the following paragraph, the presence of substantial concentrations of a wide variety of organic ligands, including siderophores, will dramatically alter the participation of iron in dark and photochemical reactions.
I suggest the authors try to add some of their insights into the manuscript, to make it more of a personal contribution to the field.
Author Response
Please see the attachment for the answers of two reviewers.

Reviewer 2 Report
In this work the authors examine the photochemistry of the aqueous phase and its effects on air pollution.
Several aspect have been correctly described and cited, but I think some are missing. For example, the role of brown compounds (see Zhao, R., et al. "Photochemical processing of aqueous atmospheric brown carbon." Atmospheric Chemistry and Physics, 15.11 (2015): 6087-6100.).
Organic chemistry is not fully described, e.g. secondary organic compounds, pynenes, humic and humic-like substances (see Herrmann, et al. "Tropospheric aqueous-phase chemistry: kinetics, mechanisms, and its coupling to a changing gas phase." Chemical reviews 115.10 (2015): 4259-4334).
I suggest publishing this work after introducing a more-in-depth discussion of these topics.
Several minor comments follow.
Line 40: you wrote " ... gases, vapours ...". I do not see the difference between gases and vapors. Please, could you be more specific about "vapours" and "gases" in this context?
Figure 1 refers to data which are not referenced. Please, add references in the caption, or introduce a table with appropriate references. Moreover, it is not clear how the medians and percentiles are calculated. Are you referring to a specific work or is it your estimate, based on literature data?
I prefer to move the Fenton reaction (line 137) just after its first quote (line 131)
Line 138: may be "it is a highly reactive and toxic substance"?
Line 154: please use kJ instead of kcal. Usually photochemical reactions need and energy close to 100 kcal
Author Response
Please see the attachment for the answers of the two reviewers.

Round 2
Reviewer 2 Report
I suggest to accept the manuscript in the present form